# The Relationship Between Camouflaging and Lifetime Depression Among Adult Autistic Males and Females

**DOI:** 10.3390/brainsci15090920

**Published:** 2025-08-26

**Authors:** Camilla Gesi, Roberta Pisani, Nicolò Tamburini, Bernardo Dell’Osso

**Affiliations:** 1Department of Mental Health and Addiction, ASST Fatebenefratelli-Sacco, Via G.B. Grassi 74, 20100 Milan, Italy; roberta.pisani@asst-fbf-sacco.it (R.P.); bernardo.dellosso@unimi.it (B.D.); 2Department of Psychiatry, Department of Biomedical and Clinical Sciences “Luigi Sacco”, University of Milan, 20100 Milan, Italy; nicolo.tamburini@unimi.it; 3“Aldo Ravelli” Center for Nanotechnology and Neurostimulation, University of Milan, 20100 Milan, Italy

**Keywords:** autism, camouflaging, depressive disorders, depression, role of sex

## Abstract

**Objective:** We aimed to investigate the relationship between camouflaging and lifetime depression among autistic people, along with the role of sex in this relationship. **Methods:** Sixty-five autistic subjects with no intellectual or language disability (34 females, 31 males), presenting to an outpatient service for the treatment of concurrent mental disorders, were administered module A of the Structured Clinical Interview for DSM-5 Disorders, the Autism Spectrum Quotient (AQ), and the Camouflaging Autistic Traits Questionnaire (CAT-Q). **Results:** No differences were found in CAT-Q total or domain scores across sexes. Subjects with lifetime depression reported significantly higher scores than those without for all CAT-Q scores, with depressed women often reporting the highest scores among the groups. The difference between depressed and non-depressed women was significant for all but the CAT-Q assimilation score. The CAT-Q total score significantly predicted lifetime depression (B = 0.053, *p* = 0.003) when controlling for age, sex, and the AQ total score. **Conclusions:** Our study expands the extant knowledge about the role of camouflaging in the mental wellbeing of autistic people by showing a correlation between camouflaging and depressive disorders throughout the lifetime among both males and females.

## 1. Introduction

Autism spectrum disorder (hereafter ‘autism’) includes a range of conditions characterized by impairments in social interaction and communication and a pattern of restricted/repetitive behaviors and interests [1]. According to the extant literature, autism affects approximately one female for every three to four males [2]. However, a growing body of literature has suggested that the gender representation within the autism spectrum might depend on the under-recognition of female forms of the disorder [2,3]. Several factors have been hypothesized to interfere with the recognition of autism among females. It has been proposed that females may both manifest core autistic dimensions to a lesser degree (e.g., poor theory of mind, scant verbal communication) [4,5] and show different behavioral manifestations than their male counterparts (e.g., mainstream, socially driven interests, stereotyped behavior latching onto eating).

### 1.1. Camouflaging in Autism

Camouflaging has also been suggested to be among the factors hampering the recognition of autism among females [6,7,8,9]. Camouflaging is conceptualized as a repertoire of behaviors and coping strategies aimed at masking autistic features in social situations to increase neurotypicality [10]. As such, subjects may strive to suppress stimming behaviors, force eye contact, mirror body language, mimic facial expressions, and rehearse conversations. These behaviors may facilitate social functioning but often come at significant emotional and psychological cost. Camouflaging affects the behavioral phenotype of autistic people, with the autistic endophenotype remaining unmodified. Autistic individuals using camouflaging may therefore appear adequate in social situations, despite a constant effort to behave ‘normally’ and a subjective struggle to fit within the neurotypical environment.

### 1.2. Impact of Camouflaging on Mental Wellbeing

Qualitative studies first revealed how camouflaging can be exhausting and associated with feelings of anxiety, stress, depression, and confusion, especially among autistic females [6,9,11]. Thus, later quantitative studies sought to investigate whether subjects engaging in camouflaging may be especially prone to poor mental health outcomes, finding various associations between self-reported camouflaging and neuroticism [12], social and general anxiety, and depressive symptoms [13], albeit with great variability in findings [13,14,15,16]. Still, it should be noted that all of the above studies enrolled subjects who self-reported autism through online surveys, while using a cross-sectional assessment of depression. The occurrence of depression across the lifetime might especially provide deeper insight into the impact of enduring camouflaging on the mental wellbeing of autistic people. Depression is in fact highly prevalent in this group, with more than 50% of subjects endorsing criteria for major depression and roughly 20% for recurrent depressive episodes across the lifespan [17]. While several factors might contribute to the high rate of depression, camouflaging is thought to enhance the emotional cost of everyday social intercourses, leading to psychological exhaustion and depressed mood and, in turn, to social withdrawal, functional impairment, reduced quality of life, and elevated risk of suicidality [18].

### 1.3. The Current Study

Based on these premises, the objective of the present research was to investigate the relationship between camouflaging and lifetime depression. We aimed to answer three main research questions: Do autistic females and males seeking treatment for concurrent mental disorders display different levels of camouflaging? Is camouflaging associated with specific dimensions of autism, and does sex play a role in these associations? Is camouflaging associated with depressive disorders, and does sex matter in this association? Therefore, we enrolled a cohort of autistic subjects with a normal IQ who were consecutively referred to a tertiary service for the diagnosis and treatment of concurrent mental disorders in autistic people and administered the Italian version of the Camouflaging Autistic Traits Questionnaire (CAT-Q), a standardized questionnaire assessing camouflaging. Moreover, since camouflaging may represent a stable pattern with effects throughout time, we used the lifetime diagnosis of depressive disorders—instead of cross-sectional depressive symptoms—to investigate the association between camouflaging and depression.

## 2. Materials and Methods

### 2.1. Sample

Participants were consecutively enrolled between January and December 2022 in the outpatient service for autistic adults of the ASST Fatebenefratelli-Sacco in Milan (Italy), a tertiary center for the diagnosis and treatment of mental comorbidities of people on the autism spectrum with no intellectual or language disability. All participants had a formal clinical diagnosis of autism spectrum disorder [19], childhood autism or Asperger’s syndrome [20], and/or autistic disorder or Asperger’s disorder, as assessed by a psychiatrist or clinical psychologist in the Italian National Health Service. This study was conducted in accordance with the Declaration of Helsinki, and all participants provided full informed consent before participating. The detailed recruitment process is provided in Figure 1. As a low-risk study, this study was approved by the Department of ASST Fatebenefratelli-Sacco of Milan as the relevant institutional review board.

### 2.2. Instruments

The assessment included both unstructured and structured interviews and self-administered questionnaires. The patients’ clinical overview and sociodemographic data were collected on a specially prepared form, using as many sources of information as available (patients’ and relatives’ recall, written clinical reports, etc.). For each participant, the inclusion diagnosis was validated by a psychiatrist experienced in the diagnostic assessment of autism.

### 2.3. Assessment of Depression

The participants were administered module A of the Structured Clinical Interview for DSM-5 Disorders (SCID-5 for DSM-5, Research Version; SCID-5-RV) [21] to assess lifetime depression. The interview was performed by a clinician with extensive experience in performing both structured and unstructured diagnostic evaluations. To the purpose of the study, subjects who received a diagnosis of past/current major depressive episode or current persistent depressive disorder were considered to be positive for lifetime depression.

### 2.4. Assessment of Autism

The assessment of autism was carried out by means of the Autism Spectrum Quotient (AQ). The AQ is a widely used questionnaire developed approximately fifteen years ago that provides a self-reported measure of autistic traits for use in adults with a normal IQ [22]. It comprises 50 questions assessing five different areas: social skill, attention switching, attention to detail, communication, and imagination. The AQ has been used as a screening tool for autism in the general population, as well as to evaluate autistic symptoms within autistic populations and other clinical groups [23,24].

### 2.5. Assessment of Camouflaging

Camouflaging was assessed using the Camouflaging Autistic Traits Questionnaire (CAT-Q) [25], a self-reported measure of social camouflaging behaviors in adults. It consists of 25 items exploring a range of behaviors that may be enacted by autistic subjects to disguise their difficulties in social contexts. CAT-Q items have been shown to load on three different factors: (1) compensation (i.e., actively compensating for difficulties in social situations), (2) masking (i.e., hiding one’s autistic features), and (3) assimilation (i.e., trying to fit in social situations). In this study, we used the recently validated Italian version of the questionnaire [26].

### 2.6. Statistical Analysis

The statistical analyses were conducted using SPSS, version 26 (International Business Machines Corporation) [27]. The demographic and clinical characteristics of autistic males and females were compared by means of the chi-squared test for categorical variables and Student’s *t*-test for continuous variables.

#### 2.6.1. Research Question 1: Do Autistic Females and Males Seeking Treatment for Concurrent Mental Disorders Display Different Levels of Camouflaging?

To this aim, CAT-Q total and factor scores were compared using Student’s *t*-test.

#### 2.6.2. Research Question 2: Is Camouflaging Associated with Specific Dimensions of Autism, and Does Sex Play a Role in These Associations?

Correlation analyses were used to investigate which dimensions of autism (i.e., the AQ domain scores) were associated with the CAT-Q total and factor scores. Linear regression was conducted, using the CAT-Q total score as the dependent variable and the AQ subscale score as the independent variable, controlling for age and sex. Correlations between the CAT-Q score and the *Communication deficit* and *Attention to details* scores of the AQ were conducted separately for males and females. Statistical tests for the difference between independent correlations among females and males were conducted using Fisher’s r-to-z transformation.

#### 2.6.3. Research Question 3: Is Camouflaging Associated with Depressive Disorders, and Does Sex Matter in This Association?

Logistic regression was used to evaluate the associations between lifetime depression (dependent variable) and age, sex, and CAT-Q and AQ total scores (independent variables). One-way ANOVA was used to compare CAT-Q scores among females and males with and without depressive disorder.

## 3. Results

The sample comprised sixty-five autistic persons (thirty-four females and thirty-one males; age range = 18–57 years). Sixty-three of the participants were White, one Asian, and one Hispanic. Their mean age at autism diagnosis was 26.5 (SD = 12.7). Sociodemographic characteristics by sex are shown in Table 1.

### 3.1. Do Autistic Females and Males Seeking Treatment for Psychiatric Comorbidities Display Different Levels of Camouflaging?

Comparisons of the CAT-Q and AQ scores between male and female autistic subjects are shown in Table 1. No differences were found in the CAT-Q total or domain scores across sexes. Higher AQ *Attention to details* and *Communication deficit* domain scores were found in females compared to males.

### 3.2. Is Camouflaging Associated with Specific ASD Dimensions? Does Sex Play a Role in These Associations?

Associations among the CAT-Q and autistic dimensions are shown in Table 2. Significant correlations at *p* < 0.05 and *p* < 0.001 are displayed. A series of linear regressions were conducted, using the CAT-Q total score as the dependent variable and the AQ subscale scores as the independent variables, controlling for age and sex. The *Communication deficit* (beta = 0.457, *p* < 0.001) and *Attention to details* (beta = 0.316, *p* = 0.015) domain scores were shown to significantly predict the CAT-Q total score. Correlations between the AQ *Communication deficit* and *Attention to details* scores and CAT-Q total score were separately assessed for males and females.

The correlations were significant among females (*Communication deficit*: r = 0.623; *p* < 0.001; *Attention to details*: r = 0.429; *p* = 0.05), but not among males (*Communication deficit*: r = 0.236; *p* = 0.279; *Attention to details*: r = 0.235; *p* = 0.280) (Figure 2). The difference between correlations independently run for females and males was significant for *Communication deficit* (Fisher’s z = 2.02, *p* = 0.04), but not for *Attention to details* (Fisher’s z = 0.91, *p* = 0.37).

### 3.3. Is Camouflaging Associated with Lifetime Diagnosis of Depressive Disorders? Does Sex Matter in This Association?

Table 3 shows a logistic regression model using lifetime depression as the dependent variable and age, sex, and the total AQ and CAT-Q scores as the independent variables. The CAT-Q score significantly predicted lifetime depression (B = 0.053, *p* = 0.003). We then compared the CAT-Q scores between subjects with and without depressive disorders, and among females and males with and without depressive disorders. As shown in Table 4, all CAT-Q scores were significantly greater for subjects with lifetime depression than for those without depression, with depressed women often reporting the highest scores among the groups (Figure 3). The difference between depressed and non-depressed women was significant for all but the CAT-Q assimilation score (all *p* < 0.001).

## 4. Discussion

In the present study, we aimed to investigate the relationship between lifetime depression and camouflaging in a sample of autistic people with no language or intellectual disability, as well as whether this relationship is affected by gender. We found a few important results that are worth discussing.

### 4.1. Levels of Camouflaging Across Sexes

We found no significant differences in the extent to which women and men on the autism spectrum exhibited camouflaging behaviors. This finding apparently contrasts with previous data [3,13,28] and with a recent meta-analysis indicating greater camouflaging in females than in males [29]. However, it should be noted that our sample included autistic subjects seeking treatment for comorbid mental disorders, which might have led to especially high levels of camouflaging, flattening the differences across sexes. According to the extant literature, camouflaging is more common among females, but it was also found to be more common in cross-sectionally evaluated individuals with greater levels of depressive symptoms [13,16]. Some scholars have also reported a significant association between late diagnosis and greater camouflaging behaviors [28], which might also be relevant to our sample.

### 4.2. Associations Between Camouflaging and Autism Dimensions, and the Role of Sex

Camouflaging was correlated with several dimensions of autism. Two domains of the AQ were significantly associated with camouflaging when controlling for gender: *Communication deficit* and *Attention to details*. Interestingly, both domains were further shown to be correlated with camouflaging in the subsample of females, but not among males, with a significant difference between males and females for the domain of *Communication deficit*. The previous literature has pointed out that autistic females may present with a subtle profile of language and communication difficulties compared to autistic males [4], mirrored in the female advantage in social motivation and social interaction found in other studies [30,31]. In contrast, our sample showed poorer self-reported communication skills in females than in men, which may explain the correlation between camouflaging and communication deficits in two different ways, not mutually exclusive. One explanation could be that the poorer communication function supported the higher level of camouflaging among women in our sample. A further hypothesis could be that females tend to overestimate their communication problems [32] due to a more demanding gender model, which promotes the use of camouflaging as a strategy to enhance social fitness. The correlation between camouflaging and attention to detail is more difficult to explain. It has been shown that performance in visuospatial attention to detail may characterize men but not women with ASD [33]. However, while affected in this cognitive domain, females might be more aware of their difficulty in inferring global meanings from social contexts and, thus, engage in camouflaging behaviors. The other way around, however, can also be reasonable, i.e., camouflaging may support a greater degree of cognitive control and sensitivity to details to achieve greater environmental monitoring. This finding is somewhat consistent with that of a previous study by Lai and colleagues [3], who reported that women (but not men) with higher camouflaging levels show better signal detection sensitivity, and that there is a close relationship between camouflaging and external monitoring skills.

### 4.3. Relationship Between Camouflaging and Lifetime Diagnosis of Depressive Disorders, and the Role of Sex

The main aim of this paper was to investigate the relationship between camouflaging and lifetime depressive disorders, positing that if camouflaging is a stable strategy that contributes to increasing the likelihood of depression, its effect may span one’s lifetime.

We found that camouflaging significantly predicts lifetime depression, when controlling for sex, age, and ASD severity, as evaluated with the AQ. Although sex was not shown to affect this relationship, comparisons among male and female subjects with or without lifetime depression showed that females with depression in their lifetime had higher mean camouflaging scores, followed by males with lifetime depression, females with no lifetime depression, and males with no lifetime depression. Except for the CAT-Q assimilation domain, camouflaging scores significantly differentiated females with depression from those without depression, while a significant difference was not found within the group of males. Previous studies investigating the impact of camouflaging on the risk of depression led to conflicting results. To the best of our knowledge, only one study has investigated the relationship between camouflaging and depression separately for autistic men and women in a sample of subjects with a clinical assessment of autism. Using a clinician-made proxy of camouflaging, Lai et al. [3] reported higher camouflaging scores among females than among males. However, higher camouflaging scores were associated with greater depressive symptoms in men, but not in women. Cage et al. [14] reported greater depressive symptoms (but not anxiety or stress) in autistic subjects self-reporting camouflaging than in those who did not. In a later study, the same group [15] reported no differences in depressive scores between subjects who constantly engaged in camouflaging and those who occasionally or inconsistently used camouflaging. Hull et al. [13] reported that self-reported camouflaging was a significant predictor of depression. Although females showed higher levels of camouflaging, there was no evidence that the relationship between camouflaging and depression was moderated by gender. Consistent with our results, Oshima and colleagues [16] found no interaction effect between camouflaging and sex in predicting depressive scores, as assessed cross-sectionally. The same study also showed a linear relationship between camouflaging scores and depression scores, while no separate analyses were performed for females and males. It is possible that the different samples of previous studies, the heterogeneity of measures used to assess camouflaging, and the cross-sectional assessment of depression led to the above conflicting results. Considering our and previous findings, future studies may profitably focus on exploiting the putative role of camouflaging in individuating autistic subjects with a greater risk of developing depressive disorders, especially among women. On the other hand, a causal relationship between camouflaging and depression among autistic people is not supported by available data. Alternative hypotheses are that concurrent depression may prompt and support camouflaging—as an attempt to cope with certain correlates of depression, such as low self-esteem or social withdrawal—or that a third factor (e.g., self-awareness and self-perception, social functioning) may influence both camouflaging and depression, explaining the association found in our study and in previous studies.

### 4.4. Limitations and Future Directions

While contributing to clarifying the relationship between camouflaging and mental health issues among autistic people, our study has a few limitations. First, whilst the use of a clinical sample—instead of a web-based sample—might be a strength of our paper, this led to a small number of subjects, limiting the statistical power of the analyses used. Second, although none of our participants had intellectual disabilities, current IQ scores were not available for all participants. Similarly, we did not include any variable assessing the overall level of functioning. The sole adult study investigating the role of IQ in camouflaging found no associations between IQ measures and camouflaging [3]; nonetheless, a confounding effect of these unassessed dimensions on our findings cannot be ruled out. Second, we assessed the presence of depressive disorders as a dichotomous variable, without considering the effect of multiple depressive episodes during one’s lifetime. Third, further mental disorders that have been shown to be associated with camouflaging, such as anxiety [13,15], were not included in the analyses. Moreover, although the overall level of autism as evaluated with the AQ did not increase the likelihood of depression over or above the effect of camouflaging, it is possible that individual cognitive or behavioral dimensions of autism interact with camouflaging, moderating its effect on depression. A possible way forward would be to investigate camouflaging in larger clinical samples while assessing a broader range of variables, such as the time since the diagnosis of autism, the number and severity of comorbid disorders, the IQ, and/or the overall level of functioning. Exploring the role of individual domains of autism in promoting camouflaging strategies could also help to identify autistic phenotypes that confer a particularly high risk for depression. Interestingly, a recent study showed that the relationship between camouflaging and depressive scores is not invariant across different cultures [16], highlighting that environmental factors not only trigger but also shape camouflaging behaviors, putatively playing an important role in the pathway from autism to depression. This should be considered when interpreting differences between males and females, as culturally determined gender roles may impress different clinical trajectories to men and women on the autism spectrum. Ideally, studies should employ both clinician observation and self-reported assessments to disentangle self-perceived, gender-based factors that may sustain both camouflaging and depression. However, a causal relationship between camouflaging and depression among autistic people has not been supported by the data thus far. Therefore, longitudinal studies are warranted to understand the directionality of the relationship between camouflaging and depression.

## 5. Conclusions

Our study corroborates and expands the extant knowledge about the role of camouflaging in the mental wellbeing of autistic people by showing a correlation between camouflaging and depressive disorders throughout one’s lifetime among both males and females. These findings support a putative role for camouflaging as a risk factor for depression and as a target for treatment. However, further studies with a longitudinal design and assessing a broader range of potentially significant variables are needed to elucidate the clinical significance of camouflaging.

## Figures and Tables

**Figure 1 brainsci-15-00920-f001:**
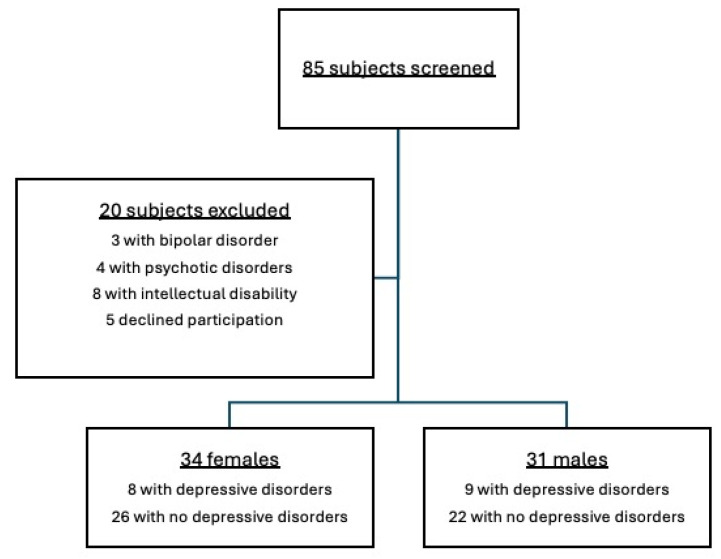
Detailed enrollment procedures and group allocation.

**Figure 2 brainsci-15-00920-f002:**
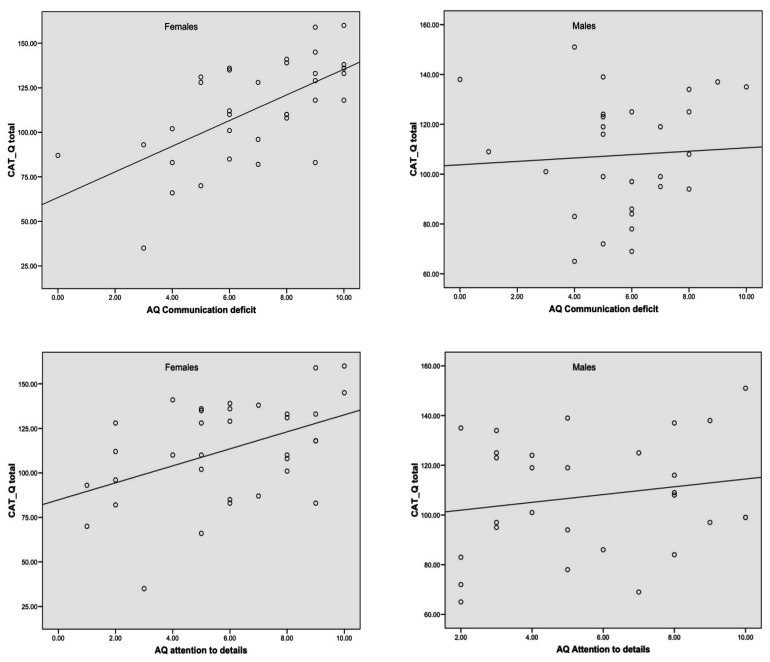
Correlations between CAT-Q total and AQ *Communication deficit* and *Attention to details* among females and males.

**Figure 3 brainsci-15-00920-f003:**
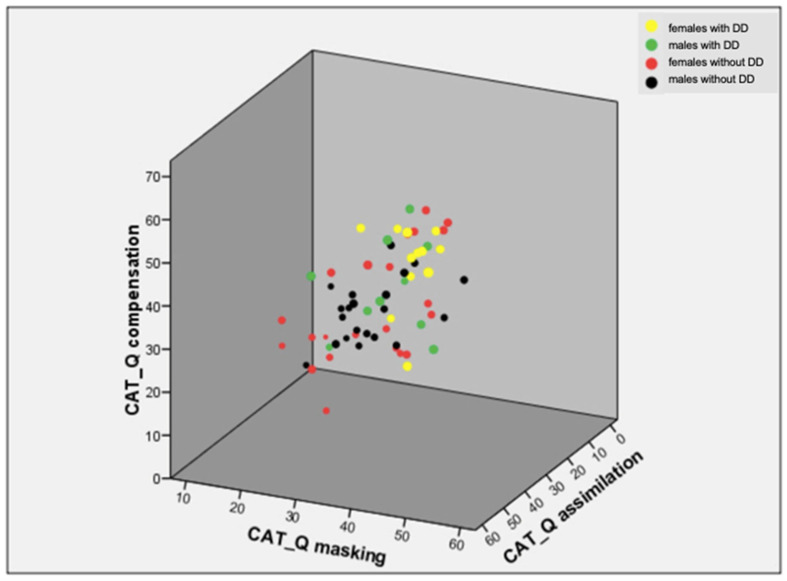
Three-dimensional (3D) scatterplot of CAT-Q subscale scores in females and males with and without depressive disorders.

**Table 1 brainsci-15-00920-t001:** Characteristics of the study sample.

*p*	Chi-Squared	Males	Females (n = 34)	
		(n = 31)		
				**Education**
0.173	4.98	4 (12.9)	8 (23.5)	Elementary
		17 (54.8)	19 (55.9)	Medium
		10 (32.3)	5 (14.7)	High school
		0 (0)	2 (5.9)	Laurea degree
**Employment Status**				
0.59	0.291	22 (71.0)	22 (64.7)	Employed
		9 (29.0)	12 (35.3)	Unemployed
**Marital Status**				
0.11	2.55	30 (96.8)	29 (85.3)	Single
		1 (3.2)	5 (14.7)	In a relationship
** *p* **	**t**			
0.589	0.544	27.5 (7.6)	28.8 (11.5)	Age
0.302	1.041	106.2 (23.3)	112.9 (28.2)	**CAT-Q total**
0.255	1.149	35.8 (10.8)	39.5 (14.9)	Compensation
0.337	0.968	34.2 (8.0)	36.4 (9.5)	Camouflage
0.703	0.383	36.2 (9.2)	37.1 (9.6)	Assimilation
0.05	2.002	28.6 (7.4)	32.6 (8.1)	**AQ total**
0.108	1.631	5.7 (2.7)	6.8 (2.7)	Social skills
**0.008**	2.739	6.9 (1.6)	8.1 (1.8)	Attention switching
**0.047**	2.029	5.7 (2.2)	6.9 (2.5)	Communication deficit
0.703	0.384	5.6 (2.7)	5.9 (2.7)	Attention to details
0.753	0.93	4.7 (2.1)	4.9 (2.0)	Imagination

**Table 2 brainsci-15-00920-t002:** Correlations among CAT-Q and AQ total and domain scores.

AQ	AQ	AQ	AQ	AQ	AQ	CAT-Q ass	CAT-Q cam	CAT-Q com	CAT-Q Total	
AD	I	C	AS	SS	Total					
										**CAT-Q** *total*
									0.894 **	**CAT-Q** *com*
								0.685 **	0.843 **	**CAT-Q** *cam*
							0.434 **	0.428 **	0.724 **	**CAT-Q** *ass*
						0.474 **	0.157	0.329 **	0.392 **	**AQ** *total*
					0.803 **	0.428 **	−0.035	0.08	0.184	**AQ** *SS*
				0.590 **	0.700 **	0.300 *	0.147	0.268 *	0.295 *	**AQ** *AS*
			0.508 **	0.628 **	0.800 **	0.465 **	0.213	0.357 **	0.423 **	**AQ** *C*
		0.366 **	0.324 **	0.489 **	0.651 **	0.308 *	−0.105	0.01	0.081	**AQ** *I*
	0.136	0.227	0.097	0.038	0.474 **	0.123	0.290 *	0.387 **	0.340 **	**AQ** *AD*

CAT-Q: com = compensation; cam = camouflaging; ass = assimilation. AQ: SS = social skills; AS = attention switching; C = communication deficit; I = imagination; AD = attention to details. * Significant at *p* < 0.05. ** Significant at *p* < 0.001.

**Table 3 brainsci-15-00920-t003:** Logistic regression for factors predictive of lifetime depressive disorders.

**Nagelkerke R2 = 0.358**	**Cox R2 = 0.251**	** *p* **	**CI 95%**	**B (SE)**	
		0.2	0.879–1.027	−051 (0.040)	**Age**
		0.003	1.018–1.093	−0.053 (0.018)	**CAT-Q total**
		0.458	0.943–1.140	0.036 (0.048)	**AQ total**
		0.807	0.207–3.409	−0.174 (0.715)	**Sex** [male]

**Table 4 brainsci-15-00920-t004:** ANOVA for comparisons among females (F) and males (M) with and without depressive disorders (DDs), respectively. Independent-samples *t*-tests were used for comparisons between overall DD and overall non-DD groups.

*p*	t	Non-DD	DD	*p*	F	F with DD	M with DD	F with No DD	M with No DD	
		n = 45	n = 20							
<0.001	−4.42	101.4 (24.4)	128.6 (18.9)	<0.001 *	10.66	134.8 (16.1)	125.2 (22.1)	101.0 (26.3)	97.2 (18.1)	CAT-Q total
0.002	−3.24	34.4 (13.0)	45.2 (10.5)	<0.001 *	6.96	49.4 (9.6)	43.1 (11.9)	34.1 (14.7)	32.3 (8.5)	CAT-Q com
0.001	−3.67	32.9 88.6)	40.9 (6.8)	0.001 *	6.38	42.9 (4.9)	38.9 (7.8)	32.8 (9.6)	32.0 (7.3)	CAT-Q cam
<0.001	−3.71	34.0 (8.5)	42.6 (8.5)	<0.001 ^§^	5.95	42.4 (7.6)	43.2 (9.7)	34.2 (9.4)	32.9 (7.0)	CAT-Q ass

* F with depressive disorder vs. M and F without depressive disorder. ^§^ F with depressive disorder vs. M without depressive disorder.

## Data Availability

The data are not publicly available due to privacy restrictions.

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
