# Peer review of "The Relationship Between Camouflaging and Lifetime Depression Among Adult Autistic Males and Females"

_brainsci, 2025, doi:10.3390/brainsci15090920_

Round 1

Reviewer 1 Report

Comments and Suggestions for Authors

This manuscript addresses an important and timely topic: the relationship between camouflaging behaviors and lifetime depressive disorders in autistic adults, with a focus on sex-related differences. The study is generally well-structured and clinically meaningful; however, several key limitations should be addressed to strengthen the manuscript:

  1. The Introduction lacks adequate depth in describing the concept and clinical relevance of camouflaging in individuals with autism spectrum disorder (ASD). To improve reader understanding, the authors should clearly define camouflaging and provide concrete examples — such as suppression of stimming behaviors, mimicking neurotypical social cues, forced eye contact, or rehearsed conversations. These behaviors may facilitate social functioning but often come at significant emotional and psychological cost. A richer conceptual framework will help contextualize the study’s rationale and its findings.
  2. The manuscript would benefit from a stronger rationale for examining lifetime depression in ASD populations. The authors should elaborate on its high prevalence and, more importantly, its serious clinical consequences, including social withdrawal, functional impairment, reduced quality of life, and elevated risk of suicidality. These risks are particularly salient in autistic individuals who camouflage, as depression in this group is often underdiagnosed or misdiagnosed. Including this context would highlight the clinical importance of the research question and the broader impact of the findings.
  3. One structural issue is the inclusion of content in the Introduction that more appropriately belongs in the Discussion — particularly the comparison of the current findings with previous literature. Since the manuscript does not fundamentally overturn prior conclusions or resolve ongoing debates, detailed interpretation and comparison of results should be reserved for the Discussion section. The Introduction should focus on presenting the current knowledge, identifying gaps, and stating the aims of the study, without preemptively analyzing findings.
  4. To improve transparency and reproducibility, I strongly recommend that the authors include a flow diagram outlining the participant recruitment process. This should include the number of individuals screened, inclusion and exclusion criteria, and final group allocation (e.g., depressed vs. non-depressed, male vs. female). At present, the sample derivation process is unclear, and such a figure would greatly enhance the methodological clarity of the manuscript.

Author Response

Comment1: The Introduction lacks adequate depth in describing the concept and clinical relevance of camouflaging in individuals with autism spectrum disorder (ASD). To improve reader understanding, the authors should clearly define camouflaging and provide concrete examples — such as suppression of stimming behaviors, mimicking neurotypical social cues, forced eye contact, or rehearsed conversations. These behaviors may facilitate social functioning but often come at significant emotional and psychological cost. A richer conceptual framework will help contextualize the study’s rationale and its findings. Answer: The reviewer made a very good point. The following sentence was included in the introduction: “As such, subjects may strive to suppress stimming behaviors, force eye contact, mirror body language and mimic facial expressions, rehearse conversations”.

Comment 2: The manuscript would benefit from a stronger rationale for examining lifetime depression in ASD populations. The authors should elaborate on its high prevalence and, more importantly, its serious clinical consequences, including social withdrawal, functional impairment, reduced quality of life, and elevated risk of suicidality. These risks are particularly salient in autistic individuals who camouflage, as depression in this group is often underdiagnosed or misdiagnosed. Including this context would highlight the clinical importance of the research question and the broader impact of the findings. Answer: We thank the reviewer for allowing us to better explain the main aim of the paper. According to his comment, the following text was included in the introduction: “Depression is in fact highly prevalent in this group, with more than 50% of subjects endorsing criteria for major depression and roughly 20% for recurrent depressive episodes across the lifespan. While several factors might contribute to the high rate of depression, camouflaging is thought to enhance the emotional cost of everyday social intercourses, leading to psychological exhaustion and depressed mood and, in turn, to social withdrawal, functional impairment, reduced quality of life, and elevated risk of suicidality

Comment 3: One structural issue is the inclusion of content in the Introduction that more appropriately belongs in the Discussion — particularly the comparison of the current findings with previous literature. Since the manuscript does not fundamentally overturn prior conclusions or resolve ongoing debates, detailed interpretation and comparison of results should be reserved for the Discussion section. The Introduction should focus on presenting the current knowledge, identifying gaps, and stating the aims of the study, without preemptively analyzing findings. Answer: again, we agreed with the reviewer and moved the paragraph about previous studies in the discussion. The discussion section was integrated mainly in the paragraph under the heading “Relationship between camouflaging and lifetime diagnosis of depressive disorders and role of sex”

Comment 4: To improve transparency and reproducibility, I strongly recommend that the authors include a flow diagram outlining the participant recruitment process. This should include the number of individuals screened, inclusion and exclusion criteria, and final group allocation (e.g., depressed vs. non-depressed, male vs. female). At present, the sample derivation process is unclear, and such a figure would greatly enhance the methodological clarity of the manuscript. Answer: a figure detailing enrollment procedures, exclusion criteria and final group allocation has been included (figure 1)

Reviewer 2 Report

Comments and Suggestions for Authors

The authors described that camouflaging is associated with females with depression disorder, but not much in male autistic patients. The camouflaging is gradually understood in ASD field from 2016 and since then, in female ASD patients, is one of important topics in female ASD patients even in net world. The authors used 2-3 different questionnaires and showed that total and domain scores in camouflaging questionnaires indicate association ASD with depression disorder. The paper is well designed and analyzed.

Minor questions.

  1. Please carefully check Fig. 1 and Fig. 2 and even in the text. For

instance, 3D presentation is Fig. 1?

  1. 3D scatterplot. Repaint walls. Because the connecting point of 3 lines in the box rises up. Should sink down.

Author Response

Comment 1: Please carefully check Fig. 1 and Fig. 2 and even in the text. For instance, 3D presentation is Fig. 1? Answer: thank you for pointing this out. The numbering was revised both in text and headings

Comment 2: 3D scatterplot. Repaint walls. Because the connecting point of 3 lines in the box rises up. Should sink down. Answer: the figure was revised as suggested

Reviewer 3 Report

Comments and Suggestions for Authors
  1. The introduction is currently presented as a single large paragraph. It is recommended to break it into smaller, logically structured paragraphs to enhance readability and clarity.
  2. Please report the age at which participants were diagnosed with ASD, as well as the age at which they were diagnosed with depression.
  3. In Section 2.6 (Statistical Analysis), it would be clearer to organize the analytic approach by hypothesis, outlining which statistical methods correspond to each specific research question.
  4. In Table 1, significant gender differences are observed in ASQ subscales such as Attention Switching and Communication Deficit. However, the miantext states that no gender differences were found.
  5. In Table 2, significance should be indicated using asterisks(*,**), rather than different colors. Additionally, tableof Linear regression results may provide clearer insights than correlation table
  6. The AdAS Spectrum questionnaire was administered but not further analyzed or discussed in later sections. Please clarify the rationale for including this measure.
  7. When comparing correlation coefficients between male and female groups, please report the statistical test for the difference between the coefficients (e.g., Fisher’s r-to-z transformation).
  8. In the logistic regression analysis, if the interaction between gender and CAT-Q scores is of interest, please include an interaction term. For group comparisons, a two-way ANOVA (i.e., 2 [gender: female vs. male] × 2 [depression vs. non-depression]) is more appropriate than a one-way ANOVA.
  9. The sample size used for correlation and regression analyses is relatively small, especially after splitting by gender (n ≈ 30 per group). Similarly, the sample size for the ANOVA is quite limited, with only 10–20 participants per cell, which may reduce statistical power.

Author Response

Comment1: The introduction is currently presented as a single large paragraph. It is recommended to break it into smaller, logically structured paragraphs to enhance readability and clarity. Answer: we found the suggestion valuable. In agreement with it, the introduction was broken down in paragraphs (Camouflaging in autism - Impact of camouflaging on mental wellbeing – Current study)

Comment 2: Please report the age at which participants were diagnosed with ASD, as well as the age at which they were diagnosed with depression. Answer: the age at diagnosis of autism was reported in the results (“Mean age at autism diagnosis was 26.5 (SD=12.7)”). Unfortunately, the age at the time of first diagnosis of depression was not collected and could not be included as suggested by the reviewer.

Comment 3: In Section 2.6 (Statistical Analysis), it would be clearer to organize the analytic approach by hypothesis, outlining which statistical methods correspond to each specific research question. Answer: the statistical methods was improved by outlining analyses used to answer to each research question.

Comment 4: In Table 1, significant gender differences are observed in ASQ subscales such as Attention Switching and Communication Deficit. However, the miantext states that no gender differences were found. Answer: the main text was integrated to account for differences shown in table 1 as follows: “Higher AQ Attention to details and Communication deficit domain scores were found in females compared to males”.  

Comment 5: In Table 2, significance should be indicated using asterisks(*,**), rather than different colors. Answer: the table was reduced by excluding AdAS Spectrum correlations. This choice (see the following point) together with the use of asterisks instead of colors, made the table rather clearer and easily readable.

Comment 6: The AdAS Spectrum questionnaire was administered but not further analyzed or discussed in later sections. Please clarify the rationale for including this measure. Answer: The reviewer is right. Our first aim was to provide a picture of associations between camouflaging and autistic dimensions as comprehensive as possible. In further analyses, to keep the paper clearer, we decided to prioritize associations with AQ dimensions only. However, following the thoughtful input of the reviewer, we thus decided to exclude the AdAS Spectrum from the paper. We think this enhanced the clarity and readability of the paper.

Comment 7: When comparing correlation coefficients between male and female groups, please report the statistical test for the difference between the coefficients (e.g., Fisher’s r-to-z transformation). Answer: the Fisher’s z’ for each correlation has been included in the results according with reviewer’s suggestion.

Comment 8: In the logistic regression analysis, if the interaction between gender and CAT-Q scores is of interest, please include an interaction term. For group comparisons, a two-way ANOVA (i.e., 2 [gender: female vs. male] × 2 [depression vs. non-depression]) is more appropriate than a one-way ANOVA. Answer: we understand the point raised by the reviewer. However, author’s aim was forst to compare subjects with and without depression, and to display the distribution of camouflaging across the four groups (men with and without depression and females with and without depression). Figure 3 was included, whilst somehow redundant, to outline this nuanced distribution.

Comment 9: The sample size used for correlation and regression analyses is relatively small, especially after splitting by gender (n ≈ 30 per group). Similarly, the sample size for the ANOVA is quite limited, with only 10–20 participants per cell, which may reduce statistical power. Answer: We agree with the reviewer. Whilst the use of a clinical sample – instead of a web-based sample – might be a strength of our paper, this led to a small number of subjects, with limited statistical power. We thus acknowledged for this limitation in the appropriate section of the manuscript.

Round 2

Reviewer 1 Report

Comments and Suggestions for Authors

The authors have addressed all the questions I raised. I recommend the manuscript for publication

Author Response

We thank the reviewer for his assistance in reviewing our manuscript

Reviewer 3 Report

Comments and Suggestions for Authors Lins 220-223: In the results section reporting gender-specific correlations, the authors present four Fisher’s z values (two for each gender) along with their p-values. However, this is misleading. If the authors aim to test whether the correlation strengths differ significantly between males and females, they should: First report the four raw correlation coefficients (r-values) for each group (i.e., males and females separately for each variable pair). Then conduct statistical tests for the difference between independent correlations using Fisher’s r-to-z transformation. This procedure would yield two Fisher’s z statistics (one per correlation pair), which test for group differences in correlation strength, rather than simply transforming each correlation into a z-value.

Author Response

Comment: Lins 220-223: In the results section reporting gender-specific correlations, the authors present four Fisher’s z values (two for each gender) along with their p-values. However, this is misleading. If the authors aim to test whether the correlation strengths differ significantly between males and females, they should: First report the four raw correlation coefficients (r-values) for each group (i.e., males and females separately for each variable pair). Then conduct statistical tests for the difference between independent correlations using Fisher’s r-to-z transformation. This procedure would yield two Fisher’s z statistics (one per correlation pair), which test for group differences in correlation strength, rather than simply transforming each correlation into a z-value.

Authors' answer: thank you for your further assistance provided on the statistics. The results section has been modified as following: "The correlations were significant among females (Communication deficit: r=.623; p<.001; Attention to details: r=.429; p=.05) but not among males (Communication deficit: r=.236; p=.279; Attention to details: r=.235; p=.280) (Figure 2). The difference between correlations independently run for females and males was significant for Communication deficit (Fisher’s z=2.02, p=.04) but not for attention to details (Fisher’s z=.91, p=.37)". Moreover, the statistical method has been integrated with the following sentence: "Statistical tests for the difference between independent correlations among females and males were conducted using Fisher’s r-to-z transformation"